# RAFT: Adapting Language Model to Domain Specific RAG

**Tianjun Zhang** [*]
Department of Computer Science
UC Berkeley
Berkeley, CA 94720, USA
{tianjunz}@berkeley.edu

**Shishir G. Patil, Naman Jain, Sheng Shen**
Department of Computer Science
UC Berkeley
Berkeley, CA 94720, USA
{shishirpatil,naman_jain,sheng.s}@berkeley.edu

**Matei Zaharia, Ion Stoica, Joseph E. Gonzalez**
Department of Computer Science
UC Berkeley
Berkeley, CA 94720, USA
{matei,istoica,jegonzal}@berkeley.edu

## Abstract

Pretraining Large Language Models (LLMs) on large corpora of textual data is now a standard paradigm. When using these LLMs for many downstream applications, it is common to additionally incorporate new information into the pretrained model either through RAG-based-prompting, or finetuning. However, the best methodology to incorporate information remains an open question. In this paper, we present Retrieval Augmented Fine Tuning (RAFT), a training recipe which improves the model's ability to answer questions in "open-book" in-domain settings. In training RAFT, given a question, and a set of retrieved documents, we train the model to ignore those documents that don't help in answering the question, which we call, distractor documents. RAFT accomplishes this by citing verbatim the right sequence from the relevant document to help answer the question. This coupled with RAFT's chain-of-thought-style response helps improve the model's ability to reason. In domain specific RAG, RAFT consistently improves the model's performance across PubMed, HotpotQA, and Gorilla datasets, presenting a post-training recipe to improve pre-trained LLMs to in-domain RAG.

## 1 Introduction

Trained on vast quantities of public data, Large Language Models LLMs have achieved significant advances in a wide range of general knowledge reasoning tasks Brown et al. (2020); Wei et al. (2022). However, increasingly LLMs are being employed in specialized domains to support tasks ranging from code completion for specific software frameworks to question answering on specific document collections (e.g., legal or medical documents). In these settings, general knowledge reasoning is less critical and instead the primary goal is to maximize accuracy based on a given set of documents. Indeed, adapting LLMs to the specialized domains (e.g., recent news, enterprise private documents, or program resources constructed after the training cutoff) is essential to many emerging applications (Vu et al., 2023; Lazaridou et al., 2022) and is the focus of this work.

This paper studies the following question – *How do we adapt pre-trained LLMs for Retrieval Augmented Generation (RAG) in specialized domains?*

When it comes to adapting LLMs to specialized domains, we consider the following two candidates: in-context learning through Retrieval-Augmented Generation (RAG) and supervised fine-tuning. RAG based methods allow the LLM to reference the documents when

---

[*]Corresponding author, personal website: `tianjunz.github.io`

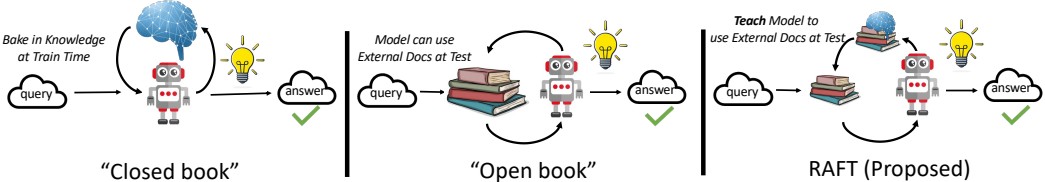

Figure 1: **How best to prepare for an Exam?**(a) Fine-tuning based approaches implement "studying" by either directly "memorizing" the input documents or answering practice QA without referencing the documents. (b) Alternatively, in-context retrieval methods fail to leverage the learning opportunity afforded by the fixed domain and are equivalent to taking an open-book exam without studying. In contrast, our approach (c) RAFT leverages fine-tuning with question-answer pairs while referencing the documents in a simulated imperfect retrieval setting — thereby effectively preparing for the open-book exam setting.

answering questions. However, RAG based in-context learning methods fail to leverage the learning opportunity afforded by the fixed domain setting and early access to the test documents. Alternatively, supervised fine-tuning offers the opportunity to learn more general patterns in the documents and better align to end tasks and user preferences Zhou et al. (2023). However, existing fine-tuning based approaches either fail to leverage the documents at test time (don't incorporate RAG) or fail to account for the imperfections in retrieval process during training.

We can draw an analogy to an open-book exam. Existing in-context retrieval methods are equivalent to taking an open-book exam without studying. Alternatively, existing fine-tuning based approaches implement "studying" by either directly "memorizing" Xiong et al. (2023) the input documents or answering practice questions Wang et al. (2022) without referencing the documents. While these approaches leverage in-domain learning they fail to prepare for the open-book nature of the test setting.

In this paper, we study how to combine instruction fine-tuning (IFT) with retrieval augmented generation (RAG). We propose a novel adaptation strategy – Retrieval-Augmented Fine Tuning (RAFT). RAFT specifically addresses the challenge of fine-tuning LLMs to both incorporate domain knowledge while also improving in-domain RAG performance. RAFT aims to not only enable models to learn domain-specific knowledge through fine-tuning, but also to ensure robustness against distracting retrieved information. This is achieved by training the models to understand the dynamics between the question (prompt), the domain-specific documents retrieved, and the right answer. Going back to our analogy to the open book exam, our approach is analogous to studying for an open-book exam by recognizing relevant, and irrelevant retrieved documents.

In RAFT, we train the model to answer the question (Q) from Document(s) (D*) to generate answer (A*), where A* includes chain-of-thought reasoning Wei et al. (2022); Anthropic (2023), and in the presence of distractor documents ($D_k$). We explain the methodology in Section 3 and analyze the sensitivity to the number of distractor documents ($k$) at train- and test- time in Section 5. RAFT consistently outperforms Supervised-finetuning both with- and without- RAG across PubMed Dernoncourt & Lee (2017), HotPot QA Yang et al. (2018), and HuggingFace Hub, Torch Hub, and Tensorflow Hub Gorilla datasets Patil et al. (2023), presenting a novel, yet simple technique to improve pre-trained LLMs for in-domain RAG. Our code is available at `https://github.com/ShishirPatil/gorilla`.

## 2 LLMs for Open-Book Exam

To understand our goal better, we expand on our analogy between training an LLM with the real-world setting of prepararing for an exam.

**Closed-Book Exam**  A closed book exam often refers to the scenario where the LLMs do not have access to any additional documents or references to answer the questions during

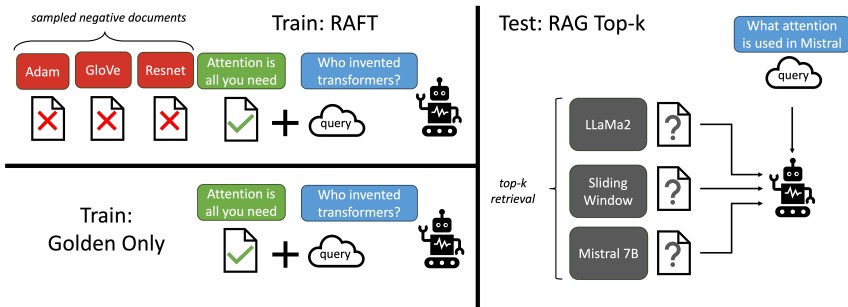

Figure 2: **Overview of our RAFT method.** The top-left figure depicts our approach of adapting LLMs to *reading* solution from a set of positive and distractor documents in contrast to standard RAG setup where models are trained based on the retriever outputs, which is a mixture of both memorization and reading. At test time, all methods follow the standard RAG setting, provided with a top-k retrieved documents in the context.

the exam. For LLMs, this is equivalent to the scenario, for example, in which the LLM is used as a chatbot. In this scenario the LLM draws from the knowledge baked in during pre-training and supervised-finetuning to respond to the users' prompt.

**Open Book Exam** In contrast, we liken the open-book exam setting to the scenario in which the LLM can refer to external sources of information (e.g., a website or a book chapter). In such scenarios, typically, the LLM is paired with retriever which retrieves 'k' documents (or specific segments of the document) which are appended to the users' prompt. It is only through these documents retrieved that the LLM gains access to "domain-specific information". As a result, we argue that the LLM's performance in these settings, where it is trained as a general-purpose LLM is largely dependent on the quality of the retriever and how accurately the retriever can identify the most relevant piece of information.

**Domain-Specific Open-Book Exam** In this paper, we focus on the narrower but increasingly popular domain than the general open book exam, which we call the domain-specific open-book exam. Here, we know apriori the domain in which the LLM will be tested. The LLM can respond to the users' prompt using use any and all information from this specific domain, which it has been fine-tuned on. Examples of domain specific examples include enterprise documents, code repositories belonging to an organization, etc. In all these scenarios, the LLM will be used to respond to the questions, whose answers can be found within a collection of documents. The retrieval technique itself has little to no-impact on the mechanism (though it may impact the accuracy). This paper studies the domain-specific open-book setting and how to adapt a pretrained LLM to this specific domain, including how to make it more robust to a varying number of retrieved documents and distractors.

## 3 RAFT

In this section, we present RAFT, a novel way of training LLMs for domain-specific open-book exams. We first introduce the classical technique of supervised fine-tuning, followed with the key takeaways from our experiments. Then, we introduce RAFT , a modified version of general instruction tuning. Lastly, we provide an overview of the experiments to expect in the later sections.

**Supervised Finetuning**

Consider the supervised fine-tuning (SFT) setting for a Question-Answer dataset. The formulation consists of the Dataset ($D$) from which a set of Question ($Q$) and corresponding answer ($A$) pairs are derived or already available. In the classical SFT setting, the model is trained to improve it's ability to answer the questions based on it's knowledge - obtained either during pre-training, or during the SFT training phase. The model so trained can also

Figure 3: RAFT prompt to help LLM evaluate its own generated reasoning and answers, contrasting them with the correct reasoning and answers. The LLM is prompted to identify errors in its reasoning and extract key insights for improvement. This figure specifically represents the 'GenerateExplanation' step in the RAFT algorithm (Section 3).

be used at test-time with Retrieval Augmented Generation (RAG) setting, where additional documents can be introduced in the prompt to help the model answer the question. This can be represented as follows:

{Train: $\mathbf{Q} \rightarrow \mathbf{A}$}, {0-shot Inference: $\mathbf{Q} \rightarrow \mathbf{A}$}, {RAG Inference: $\mathbf{Q} + \mathbf{D} \rightarrow \mathbf{A}$}

**RAFT:** Retrieval Augmented Fine-Tuning (RAFT), presents a novel recipe to prepare fine-tuning data to tailor the models for domain-specific open-book setting, equivalent to in-domain RAG In RAFT, we prepare the training data such that each data point contains a question ($Q$), a set of documents ($D_k$), and a corresponding Chain-of-though style answer ($A^*$) generated from one of the document ($D^*$). We differentiate between two types of documents: 'golden' documents ($D*$) i.e. the documents from which the answer to the question can be deduced, and 'distractor' documents ($D_i$) that do not contain answer-relevant information. As an implementation detail, the 'golden' document doesn't need to be a single document, but can be more than one document, as is the case in HotpotQA Yang et al. (2018). Then, for $P$ fraction of the questions ($q_i$) in the dataset, we retain the golden document ($d_i^*$) along with distractor documents ($d_{k-1}$). For ($1-P$) fraction of the questions ($q_i$) in the dataset, we include no golden document and only include distractor documents ($d_k$). We then fine-tune the language model using standard supervised training (SFT) technique, training it to generate answers from the provided documents and question. Fig. 2 illustrates the high-level design principal for RAFT .

We demonstrate that our RAG approach trains the model to perform better RAG on the set of documents it is trained on *i.e., in-domain*. By removing the golden documents in some instances, we are compelling the model to memorize answers instead of deriving them from the context. The training data for RAFT is as follows, and an example training data can be seen in Fig. 3:

**P** % of data: $\mathbf{Q} + \mathbf{D}^* + \mathbf{D}_1 + \mathbf{D}_2 + \ldots + \mathbf{D}_k \rightarrow \mathbf{A}*$

(1 − **P**) % of data: $\mathbf{Q} + \mathbf{D}_1 + \mathbf{D}_2 + \ldots + \mathbf{D}_k \rightarrow \mathbf{A}*$

Subsequently, for the test scenario, the model is provided with the Q and top-k documents retrieved by the RAG pipeline. Note that RAFT is independent of the retriever used.

A key factor in enhancing training quality is the generation of a reasoning process, such as Chain-of-Thought, to explain the provided answers. RAFT  approach is similar: we demonstrate that creating a full reasoning chain and in-addition, clearly citing sources enhances the model's accuracy in answering questions. In Fig. 3, we illustrate this set-up. Generating the training data in this fashion, involves presenting the model with a question, context, and verified answers, and then requesting it to form a reasoning chain that appropriately references the original context.

For all the datasets in our experiments, we generate the answers using the technique described above. Note that the Gorilla APIBench dataset, already includes reasoning in the answers. We provide an example of the generation step in Fig. 3, the detailed reasoning answer includes a citation from the original context inside `##begin_quote##` and `##end_quote##` as well as the detailed explanation on how to reach the conclusion based on the citations. We demonstrate that adding detailed reasoning paragraphs can help boost the model's performance in our experiment section.

Table 1: **RAFT improves RAG performance for all specialized domains**: Across PubMed, HotPot, HuggingFace, Torch Hub, and Tensorflow Hub, we see that Domain-specific Fine-tuning improves significantly of the performance of the base model, RAFT consistently outperforms the existing domain-specific finetuning method with or without RAG. This suggests the need to train the model with context. We compare our model with LLaMA finetuning receipes, and provide GPT-3.5 for reference.

|  | PubMed | HotPot | HuggingFace | Torch Hub | TensorFlow |
|---|---|---|---|---|---|
| GPT-3.5 + RAG | 71.60 | **41.5** | 29.08 | 60.21 | 65.59 |
| LLaMA2-7B | 56.5 | 0.54 | 0.22 | 0 | 0 |
| LLaMA2-7B + RAG | 58.8 | 0.03 | 26.43 | 08.60 | 43.06 |
| DSF | 59.7 | 6.38 | 61.06 | 84.94 | 86.56 |
| DSF + RAG | 71.6 | 4.41 | 42.59 | 82.80 | 60.29 |
| RAFT (LLaMA2-7B) | **73.30** | 35.28 | **74.00** | **84.95** | **86.86** |

# 4    Evaluation

We design our experiments to study how well RAFT performs compared to various baselines. We find that the RAFT-7B model (a finetuned version of LlaMA-2) is better at reading and extracting information from in-domain documents, than domain-specific finetuned model, and general-purpose model with RAG. As an ablation, we also demonstrate how important it is for the model to learn with Chain-of-Thought responses. In this section, we will first introduce all the datasets we used in the experiments, then all the baseline model/fine-tuning techniques that we benchmark against.

**Datasets**    In our experiments, we use the following datasets to evaluate our model and all baselines. We selected these datasets to represent both popular and diverse domains including Wikipedia, Coding/API documents, and question-answering on medical documents. Natural Questions (NQ) Kwiatkowski et al. (2019), Trivia QA Joshi et al. (2017) and HotpotQA Yang et al. (2018) are the open-domain question-answers based on Wikipedia, mainly focused on common knowledge (e.g., movies, sports, etc). HuggingFace, Torch Hub, and TensorFlow Hub are from the APIBench Patil et al. (2023) proposed in the Gorilla paper. These benchmarks measure how to generate the correct, functional, and executable API calls based on the documentation. PubMed QA Jin et al. (2019) is a question-answering dataset tailored only for biomedical-research question-answering. It mainly focuses on answering medical and biology questions based on a given set of documents. We would like to highlight that (NQ, Trivia QA, and HotpotQA) are relatively general domain whereas the latter two domains are on domain-specific documents.

**Baselines**    We consider the following baselines for our experiments:

- LlaMA2-7B-chat model with 0-shot prompting: this is the commonly used instruction-finetuned model for QA tasks, where we provide clearly written instructions, but no reference documentation.

- LlaMA2-7B-chat model with RAG (Llama2 + RAG): similar to the previous setting, except here we include reference documents. This is a popular technique when dealing with domain-specific QA tasks.

- Domain-Specific Finetuning with 0-shot prompting (DSF): Standard supervised-finetuning, without documents in context. We find that its mostly useful to align the answering style of the model as well as get familiar with the domain context.

- Domain-Specific Finetuning with RAG (DSF + RAG): Equip a domain-specific finetuned-model with external knowledge using RAG. So, for the "knowledge" the model does not know, it can still refer to the context.

Table 2: **Ablation on Chain-of-Thought**: The numbers of RAFT and RAFT without CoT. Results on various datasets show that adding CoT can significantly improve the performance of the finetuned model. With a gains of 9.66% and 14.93% in the Hotpot QA and HuggingFace datasets respectively.

|              | PubMed    | HotpotQA  | HuggingFace | Torch Hub | TensorFlow |
|--------------|-----------|-----------|-------------|-----------|------------|
| RAFT w.o CoT | 68.30     | 25.62     | 59.07       | **86.56** | 83.21      |
| RAFT         | **73.30** | **35.28** | **74.00**   | 84.95     | **86.86**  |

## 4.1 Results

Using the above datasets and baselines, we evaluate our model RAFT and demonstrate the effectiveness of RAFT in Tab. 1. We see that RAFT consistently and significantly outperforms the baselines. Compared with the base Llama-2 instruction-tuned model, RAFT with RAG does much better in terms of extracting information as well as being robust towards distractors. The gain can be as big as 35.25% on Hotpot QA and 76.35% on Torch Hub evaluation. Compared with DSF on the specific dataset, our model does better at relying on the provided context to solve the problem. RAFT does much better on the tasks like Hotpot and HuggingFace datasets (30.87% on Hotpot and 31.41% on HuggingFace). Note that for PubMed QA, since it is a binary yes/no question, we don't observe significant gains when we compare our model with DSF + RAG. Even compared with a much larger and better model GPT-3.5, RAFT demonstrates significant advantages.

Overall, the LLaMA-7B model, both with and without the RAG, performs poorly due to its answering style not aligning with the ground truth. By applying domain-specific tuning, we significantly enhance its performance. This process enables the model to learn and adopt the appropriate style of answering. However, introducing RAG to a domain-specifically fine-tuned (DSF) model doesn't invariably lead to better outcomes. This might indicate that the model lacks training in context processing and extracting useful information from it. By incorporating our method, RAFT , we train the model not only to match its answering style with that required but also to improve its document processing capabilities. Consequently, our approach outperforms all others.

## 4.2 Effect of CoT

We also conduct an analysis to evaluate the effectiveness of the Chain-of-Thought approach in enhancing the model's performance. As indicated in Table 2, simply providing the answer to a question may not always be adequate. This approach can lead to a rapid decrease in loss, resulting in the model beginning to overfit. Incorporating a reasoning chain that not only guides the model to the answer but also enriches the model's understanding can improve the overall accuracy and prevent overfitting to concise answers. In our experiments, integrating the Chain-of-Thought significantly enhances training robustness. We employ GPT-4-1106 to generate our Chain-of-Thought prompts and include an example of the prompt we used in Figure 3.

## 4.3 Qualitative Analysis

To illustrate the potential advantages of RAFT over the domain-specifically fine-tuned (DSF) approach, we present a comparative example in Figure 4. This example qualitatively demonstrates a scenario where the DSF model becomes confused by a question asking for the identity of a screenwriter. Instead of providing the correct name, it mistakenly cites one of the films written by the screenwriter. In contrast, the RAFT model accurately answers the question. This discrepancy suggests that training a model solely with question-answer pairs may impair its ability to derive relevant context from provided documents. The comparison underscores the importance of incorporating both standard instructional tuning and context comprehension into the training dataset to preserve and enhance the model's ability to process text effectively.

---

**HotPot QA**

**Question:** What screenwriter with credits for "Evolution"[0/1879]e a film starring Nicolas Cage and Téa Leoni?

**Documents:** ... David Weissman is a screenwriter and director. His film credits include "The Family Man" (2000), "Evolution" (2001), and "When in Rome" (2010). ...

The Family Man is a 2000 American romantic comedy-drama film directed by Brett Ratner, written by David Diamond and David Weissman, and starring Nicolas Cage and Téa Leoni. ...

______________________

**RAFT:** ##Reason: The screenwriter with credits for the film "Evolution," starring Nicolas Cage and Téa Leoni, is David Weissman. This information is provided in the reference documents where it mentions David Weissman as a screenwriter with film credits including "The Family Man" (2000), "Evolution" (2001), and "When in Rome" (2010). Therefore, the screenwriter for "Evolution" is David Weissman. ##Answer: David Weissman ✓

**DSF:** The Family Man ✗

---

Figure 4: **Comparison of RAFT and DSF**: On the HotPot QA dataset, we can see that DSF model extracts the wrong information from the context when the question is asking who is the screen writer and it answers a film name. RAFT manages to get the accurate results .

### 4.4 Should we train the LLM always with the golden context for RAG?

In our exploration of whether large language models (LLMs) should always be trained with the golden context for Retrieval-Augmented Generation (RAG), we address a key question: what proportion (p%) of the training data should include golden documents? Intuitively, one might assume that for effective training in reading and extracting information from context (e.g., RAG tasks), the golden document should always be included during training (P = 100%). However, our findings challenge this assumption: incorporating a portion of the training data without the golden document in the context (P = 80%) appears to enhance the model's performance on RAG tasks.

Figure 5 presents our investigation into the hyperparameter P%, which represents the percentage of training instances that should include golden documents. We find that the optimal proportion varies across datasets, with P% ranging from 40%, 60%, and 100%. This indicates that training your LLM without the correct corresponding context at times can be beneficial for the downstream task of answering questions related to the documents. In our training setup, we include four distractor documents alongside the golden document, and at test time, we maintain this format by providing the golden document with four distractors. Our findings suggest that, for domain-specific RAG tasks, including a certain percentage of training data without the golden documents in the context proves to be advantageous.

## 5 RAFT Generalizes to Top-K RAG

We now study another important problem: How does the number of distractor documents in RAFT affect the model's performance when augmented with top-k RAG results during evaluation? Previous research has highlighted the vulnerability of LLMs to irrelevant text (see studies (Shi et al., 2023a; Weston & Sukhbaatar, 2023; Liu et al., 2023)). This issue is particularly critical for LLMs + RAG since top-k RAG is frequently employed at test time to ensure high recall. Such a scenario necessitates the model to have the ability to discern and disregard irrelevant content, focusing solely on pertinent information.

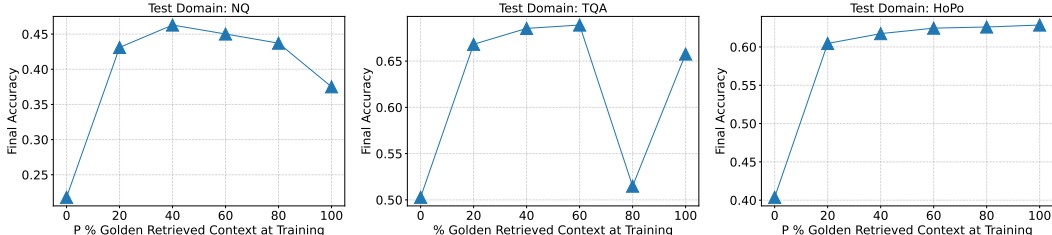

Figure 5: **How many golden documents to involve?** We study the hyperparameter P% where it indicates how much portion of training data is with golden document. Results on NQ, TQA and HotpotQA suggest that mixing some amount of data that the golden document is not put in the context is helpful for in-domain RAG.

## 5.1 Making Model Robust to top-K RAG

To tackle the challenge of enhancing large language models' (LLMs) ability to sift through irrelevant text within the retrieval pipeline, our analysis revealed that training solely with golden (highly relevant) documents can inadvertently diminish the model's ability to discern and disregard irrelevant information. To address this, our algorithm, RAFT , adopts a strategy that integrates golden documents with a mix of irrelevant ones. This methodology prompts us to investigate the ideal fraction of distractor (irrelevant) documents to incorporate throughout the training process and to assess how well this training approach adapts to different volumes of documents encountered by the Retrieval-Augmented Generation (RAG) during the test phase. Our aim is to refine the balance between relevant and irrelevant information to strenghten the model's efficiency in identifying and utilizing pertinent content. Notice that Sec 4.4 looked what what P% of training data should include distractors, while in this section, we study test-time scenarios.

**Training with Distractor Documents** To enhance the robustness of LLMs against irrelevant text in retrieved documents, we adopted a finetuning approach that incorporates both golden (highly relevant) documents and distractor (irrelevant) documents. The model was trained with varying numbers of distractor documents, but consistently evaluated using the top-3 documents obtained from the retriever - not to be confused with $p$. Our findings, detailed in Fig. 6, reveal that finetuning with only the golden document frequently results in inferior performance compared to configurations that include a greater number of distractor documents. As we can see in the figure, the better performance for Natural Questions is training with $D^* + 3D$ and it is $D^* + 1D$ documents with Hotpot QA. This insight has been particularly beneficial for our algorithm, RAFT . In our experiments, we consistently employ a training setup consisting of one golden document alongside four distractor documents.

**Generalization to a variable number of test-time documents.** We extended our research to examine the impact of different quantities of test-time documents on the model's performance. Specifically, our experiments focused on assessing how models, trained with varying numbers of distractor documents, respond to changes in the number of documents presented at test time. The results, illustrated in Fig. 6, confirm that the inclusion of distractor documents during training indeed makes the model more resilient to fluctuations in the number of documents encountered during testing. This ability to maintain consistent performance despite variations in test-time document numbers further validates the robustness of our approach, RAFT . This finding underscores the importance of a well-calibrated training environment to prepare the model for a range of scenarios it may encounter in real-world.

## 6 Related Works

**Retrieval-Augmented Language Models** Retrieval-Augmented Language Models (RALMs) enhance LLMs by integrating a retrieval module that sources relevant information from external knowledge bases, significantly improving performance across various NLP tasks, including language modeling (Guu et al., 2020; Borgeaud et al., 2022; Khandelwal et al.,

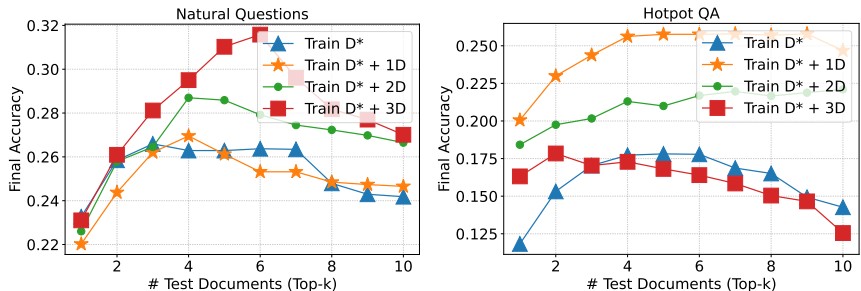

Figure 6: **Test-Time Documents Varying**: To analyze how robust RAFT is to varying number of test-time documents, we study three domains – NQ, Trivia QA and HotPot QA. In NQ, we find that training with 4 documents leads to optimal performance, and this changes to 3 and 2 for for Trivia QA and HotPot QA respectively. However, we see that training with only *golden* documents leads to poor performance.

2019; Shi et al., 2023d; Lin et al., 2023b; Shi et al., 2023c; Asai et al., 2023; Xu et al., 2023; Wang et al., 2023) and open-domain question answering (Izacard et al., 2023; Lewis et al., 2020). For instance, Atlas (Izacard et al., 2023) fine-tunes T5 models with the retriever, treating documents as latent variables, while RETRO (Borgeaud et al., 2022) modifies the decoder-only architecture to include retrieved texts and conducts pre-training from scratch. kNN-LM (Khandelwal et al., 2019) interpolates between the LM's next token distribution and distributions computed from retrieved tokens at inference. (Shi et al., 2023d; Ram et al., 2023) assume black-box access to an LLM, combining it with either off-the-shelf or fine-tuned retriever.

**Memorization** A key question around large neural language models is whether they truly "understand" text (Feldman, 2020; Power et al., 2022) or simply rely on surface pattern memorization (Carlini et al., 2019; Tänzer et al., 2022). (Feldman, 2020; Carlini et al., 2019; 2022) develop methodologies to quantify the extent of memorization in neural models. (Brown et al., 2020; Power et al., 2022; Liu et al., 2022) further explored how memorization impacts the models' generalization capabilities. (Carlini et al., 2021; Shi et al., 2023b) demonstrated the ability of language models to memorize and regurgitate training data, raising significant privacy concerns (Kandpal et al., 2022; Pan et al., 2020).

**Finetuning for RAG** More recently, several papers have been exploring the idea of fine-tuning a pretrained LLM to be better at RAG tasks (Lin et al., 2023a; Wang et al., 2023; Xu et al., 2023; Liu et al., 2024). These works focus on constructing a combination of finetuning dataset for RAG and train a model to perform well on these tasks. In particular, in their settings, at test time, the domain or documents can be different than the training time; whereas our paper studies a slightly opposite scenario where we only care about testing the LLM on the same set of documents.

## 7 Conclusion

RAFT is a training strategy designed to enhance the model's performance in answering questions within a specific domain, in "open-book" settings. We highlight several crucial design decisions, such as training the model alongside distractor documents, organizing the dataset so a portion lacks golden documents in their context, and formulating answers in a chain-of-thought manner with direct quotations from the relevant text. Our evaluations on PubMed, HotpotQA, and Gorilla API Bench underline RAFT's significant potential.

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
