# OpenReview forum: "RAFT: Adapting Language Model to Domain Specific RAG"
_colmweb.org/COLM/2024/Conference — COLM_

### Official Review · Reviewer_2uZs · 2024-05-02

**Rating:** 6
**Confidence:** 4
**Ethics Flag:** 1

**Summary:**

This paper introduces RAFT to improve LLM's RAG performance. It does this by adding distractive documents and chain-of-thought reasoning during fine-tuning on open-book QA datasets. Including distractive documents helps the LLM focus on important info and ignore unimportant details in the top-k documents. Also, adding chain-of-thought reasoning makes its responses more accurate.

**Questions To Authors:**

1. Regarding Chain-of-Thought, I'm curious about whether fine-tuning using CoT would affect the performance of tasks not originally designed as CoT. For instance, what about the ablations on NQ and TQA? I suspect that the fine-tuned model might initially generate some CoT before producing the final answer. Therefore, I'm interested in whether the final answer aligns with the ground truth.

2. Can the domain-specific knowledge learned from the trained dataset be applied to other types of tasks? For instance, could a model trained on PubmedQA also perform well on benchmarks like MedMCQA [1] or Bio-MMLU [2]  also in the biomedicine domain? As suggested in [3], models trained on domain-specific reading comprehension corpora (i.e., a form of open-book QA) can apply domain-specific knowledge to other tasks within the same domain, which is similar to your fine-tuning setting.

[1] MedMCQA: A Large-scale Multi-Subject Multi-Choice Dataset for Medical domain Question Answering

[2] Measuring Massive Multitask Language Understanding

[3] Adapting Large Language Models via Reading Comprehension

**Reasons To Accept:**

The two methods proposed in the paper, namely, incorporating distractive documents into the context and implementing reasoning chains in the models, are intuitive and have demonstrated significant enhancements in the model's performance across various benchmarks.

The paper provides thorough experiments to support its assertions.

**Reasons To Reject:**

1. Applying Chain-of-Thought (CoT) to all the fine-tuning tasks may introduce limitations.

While CoT is known to be helpful for challenging questions, it might not be suitable for simpler tasks, such as sentiment analysis. I notice that in Table 1 and Table 2, the authors only present main experiments and ablations on tasks originally designed as explainable tasks (i.e., tasks requiring CoT). However, what about the main results and ablations for NQ and Trivia QA, which were not originally presented in the CoT mode?

Also, Table 1 exceeds the page width.

2. The evaluation of domain-specific knowledge is not sufficient.

While the author proposes that RAFT can inject domain-specific knowledge into the LLM, it's essential to understand if the fine-tuned LLM can effectively apply its acquired domain knowledge from seen datasets to unseen datasets within the same domain.

For example, consider PubmedQA, which enriches the LLM with biomedical knowledge. During testing, can the fine-tuned model perform well on task types not directly related to PubmedQA but could still benefit from the domain knowledge learned from the PubmedQA dataset?

---

> ### Author Rebuttal · Authors · 2024-05-31
>
> We would like to first thank the reviewer for their time and helpful feedback on our paper. We will address the reviewer’s comments in detail below.
>
> 1. Chain of thought on NQ and TQA:
> CoT in general can help performance even in those scenarios that were not originally designed to be CoT. This is because the model can provide more context to pick and express the right answer. We will provide one example here, in one of NQ questions, “What is the last movie of Jason Bourne” the right answer as per the dataset is “Jason Bourne”. With CoT, the LLM can answer “Jason Bourne(2016)” by referencing the documentation within its context. We will add additional NQ and TQA ablation similar to the HotPot-QA ablation to the paper in the next revision.
>
> 2. Insufficiency of Domain-Specific Knowledge: The reviewer raised a valid point. We want to clarify that in our experiments, we include the following two settings:
>
> (a) In TorchHub, TensorHub, and HuggingFace, our test questions are based on the same documents as the training set. This would mean that we train our question-document-answer triplets on document set A, and then use question-document from the same document set A for testing.
>
> (b) In Hotpot QA, NQ, TQA and MedPub, our testing questions don’t necessarily refer to the same documents in the training set. This would mean that we train our question-document-answer triplets on document set A, and then use question-document from document set B for testing.
>
> As shown in the results (Table. 1 and Figure. 5), RAFT can successfully generalize to questions regarding similar domains with documents that are not seen in the training set. We will add more evaluations to understand the performance of RAFT.

---

> > ### Comment · Reviewer_2uZs · 2024-06-03
> > **Reply to authors**
> >
> > Dear Authors,
> >
> > **Regarding the CoT Setting**
> >
> > I hope you can show the results for NQ and TQA because running an inference and reporting the results does not take too much time, as far as I know. Moreover, from Figure 6 in your paper, you've already done some experiments on NQ and TQA, which suggests it is not too hard for you to run and report these results.
> >
> > **Regarding Domain Knowledge**
> >
> > Cross-document evaluation (where the documents are still from the same dataset) may not be enough to demonstrate that the model has gained generalizable domain knowledge. I highly recommend adding at least one cross-dataset experiment to strengthen your claims.

---

> > ### Author Response · Authors · 2024-06-06
> > **Author Response**
> >
> > We would like to thank the reviewer for spending time replying to us. We have some new results below:
> >
> > **CoT Setting**: We ran some additional ablation on the CoT for NQ and TQA and reported the results below. As the reviewer mentioned, we see a slight degradation of CoT compared to non-CoT settings (but roughly the same number). This might be because the TQA and NQ tasks focus on citing the answering text from the context above without much reasoning involved. We want to say that given the results are very similar, adding CoT is still in general helpful in the settings where the reasoning is heavily involved. However, this indeed is an interesting finding, we will clarify this in the paper and add a discussion on this.
> >
> > NQ: CoT: 49.10%; Non-CoT: 50.90%
> > TQA: CoT: 62.77%; Non-CoT: 64.77%
> >
> > **Domain Knowledge**
> > We managed to run some ablation on how the knowledge can be generalized between datasets. We have three anonymous figures here. In this setting, we train on NQ/TQA/Hotpot and test on News QA/Squad QA/Web Questions. We found that RAFT can indeed generalize to new domains as well. In the figure, we mainly study how the percentage of the golden documents ($p$%) included in the training dataset will affect the generalization performance. The curves demonstrate that domain generalization can mostly benefit from a training dataset of 100% golden documents, as remembering the knowledge in one domain won't necessarily help another.
> >
> > https://drive.google.com/file/d/1R30PcDaMeMyady3NZ6uFj-vlnHYron6i/view?usp=sharing
> >
> > https://drive.google.com/file/d/1dckLBNICJL4YCDvkyoYc4euQFCHGtgGD/view?usp=sharing
> >
> > https://drive.google.com/file/d/1bM860jzMlwmt9UqWgTEO7wam0fkxoK7A/view?usp=sharing

---

> > > ### Comment · Reviewer_2uZs · 2024-06-06
> > > **Reply to authors**
> > >
> > > Thanks for the reply, it resolves my confusion and my review is still positive.

---

### Official Review · Reviewer_Zx57 · 2024-05-06

**Rating:** 6
**Confidence:** 5
**Ethics Flag:** 1

**Summary:**

This paper proposes Retrieval Augmented Fine Tuning (RAFT), a fine-tuning method leveraging RAG in domain-specific question-answering tasks. The main idea is to fine-tune LLMs on target domain data with a combination of oracle document context and distractor document context. Comparing with raw LLM, LLM with RAG, and Finetuned LLM, RAFT significantly performs the baselines. Analysis on the fine-tuning data is shown to demonstrate how RAFT works.

**Questions To Authors:**

One possible way to justify RAFT is effective is to apply it to tasks other than open-domain QA, i.e. when the domain-specific annotation is unknown, or when the retrieval is not an essential step. Do you consider trying tasks such as text-to-SQL or reasoning-based QA?

**Reasons To Accept:**

1. The proposed method achieves strong performance compared to baselines.
2. Useful results to prepare domain-specific fine-tuning data in a specific domain, including various baselines and datasets.

**Reasons To Reject:**

The idea is relatively straightforward and lacks novel contribution. Domain-specific fine-tuning and RAG are widely known to be effective to solve domain-specific tasks. Both of them are ways to leverage the domain training data. Thus, the results are mostly expected in this paper. Compared to baselines, the proposed method has the following advantages, retrieval and domain-specific annotation:
1. Retrieval is an essential step in the open domain QA task and DSF without proper prompting is a meaningless baseline. DSF with question-answer pairs could only lead to memorization of answers for LLMs since there is no clue how these answers can be generated from the input.
2. Domain-specific annotation is also important. Even though the LLMs can generalize well on most common knowledge, they still struggle on domain-specific problems and need the downstream task's annotation. Thus, comparing RAFT to baselined without domain-specific fine-tuning, the performance gain is also expected.
Thus, combining these two components is not a novel work.

---

> ### Author Rebuttal · Authors · 2024-05-31
>
> We would like to first thank the reviewer for their time and helpful feedback on our paper. We will address the reviewer’s comments in detail below.
>
> 1. Combing RAG and finetuning naturally lead to improvement:
>
> Paper contribution: In our paper, we study the setting of domain-specific “open-book” exams, where we know apriori the domain in which the LLM will be tested. Both RAG and Finetuning are popular approaches for this task, so it seems natural to combine them. But to the best of our knowledge, RAFT studies the effect of combining both thoroughly and demonstrates one can finetune a much smaller model (Llama2-7B) to achieve comparable performance to GPT-3.5. We identify techniques like the verbatim chain-of-thought, distractor documents, and percentage of golden documents, deriving a good methodology for the domain-specific RAG field.
>
> Baselines in our experiments: Our experiments study why RAFT is improving over baselines. There are two potential reasons: (1) improved understanding and reasoning on the given domain and (2) improved answering styles that are easier for the exact matching methods to check. That’s why we choose RAG and DSF as baselines. In Table. 1, we first compare RAFT to Llama2-7B-instruct + RAG, and GPT3.5 + RAG, which serve as one solution to adapt general LLM models to domain-specific QA tasks through injecting domain-specific knowledge to LLMs. Note that this baseline even uses a much stronger base model GPT3.5, compared to RAFT uses Llama2. Second, the main reason we compare RAFT to DSF is to understand the effect of answering style improvement. Using DSF, the model is finetuned to provide the same answering style that is the same as the RAFT model, with the only difference to memorize the knowledge. Thus, this illustrates RAFT does improve the model’s capability of understanding and reasoning over the given context.
>
> 2. RAFT on text-to-SQL or reasoning-based QA: Our paper mainly studies the setting of domain-specific RAG. RAFT is proposed and specifically designed to boost the base LLM’s performance on this specific task. Although it may introduce performance boosts in the coding (text-to-SQL) or reasoning (reasoning QA) domain, it is out of the paper’s scope to study the performance of RAFT on those domains, since they belong to the general post-training of LLMs. We view RAFT as a method to create domain expert RAG models rather than a general post-training strategy that tries to improve the model’s general performance.

---

> > ### Comment · Reviewer_Zx57 · 2024-06-05
> >
> > Hi, Authors,
> >
> > Thanks for the reply. I acknowledge the solid results for the proposed domain expert RAG models. Although it might not be included in this paper's scoop, I expect it to be expanded to include more NLP tasks and domains.
> >
> > Given the strong results, I am okay with accepting the paper.
> >
> > Missing reference: Distant Supervision for Multi-Stage Fine-Tuning in Retrieval-Based Question Answering
> >  https://dl.acm.org/doi/abs/10.1145/3366423.3380060

---

> > > ### Author Response · Authors · 2024-06-06
> > > **Thank you for your reply**
> > >
> > > We would like to thank the reviewer for spending time replying to us. We'll add the missing reference in our next revision.

---

### Official Review · Reviewer_HVHi · 2024-05-11

**Rating:** 9
**Confidence:** 4
**Ethics Flag:** 1

**Summary:**

The authors introduce retrieval-augmented finetuning (RAFT), which trains LLMs to be more robust to retrieval errors in open-domain QA by including distractor documents during instruction fine-tuning. This is achieved by incorporating training examples with mixtures of oracle and distractor documents and without oracle documents entirely, as well as the full reasoning chain to identifying the answer. The authors demonstrate consistent performance improvements over standard RAG and finetuning on PubMed, HotpotQA, and Gorilla APIBench.

**Reasons To Accept:**

- The authors introduce and demonstrate RAFT as an effective training strategy for RAG-style tasks. Introducing distractor documents during instruction fine-tuning improves these models' ability at inference time.
- There is a clear hypothesis, good experimental design, and reasonable ablations
- Results are provided over a good range of datasets and demonstrate the effectiveness of the technique
- The paper is well written and organized

**Reasons To Reject:**

No obvious reasons

---

> ### Author Rebuttal · Authors · 2024-05-31
>
> We would like to first thank the reviewer for their time and helpful feedback on our paper. We look forward to discussing the paper with the reviewer in the discussion phase if the reviewer has further concerns.

---

### Official Review · Reviewer_RbkE · 2024-05-11

**Rating:** 7
**Confidence:** 5
**Ethics Flag:** 1

**Summary:**

This paper presents RAFT, a fine-tuning method designed to enhance the performance of retrieval-augmented generation (RAG) for language models in open-book, in-domain question answering scenarios. The approach involves training language models using a mix of oracle and distractor documents, enabling them to discern relevant information from less useful content in the top-k retrieved documents. By comparing their method against conventional RAG baselines, the authors demonstrate that their approach significantly outperforms existing methods across five question answering datasets.

**Reasons To Accept:**

[1] The RAFT approach is not only straightforward to implement but also consistently outperforms baseline models by a significant margin across a variety of domain-specific question answering datasets in the open-book setting.

[2] The paper is well-organized and includes figures and examples that make the central concepts more intuitive and easier to understand.

[3] In addition to using oracle documents during training, the authors effectively incorporate distractor documents and systematically demonstrate their impact on model performance.

**Reasons To Reject:**

[1] Data Usage Inconsistency: The authors seem to focus on domain-specific datasets such as PubMed and Huggingface, as indicated in the title. However, they have not explicitly stated the inclusion of general domain datasets like NQ and TQA in Section 5. This data usage inconsistency may lead readers to question the analysis of "domain-specific" settings.

[2] Term Inconsistency: There are several key terms used inconsistently throughout the paper, which could reduce readability and clarity. The inconsistencies include:

Is it 'Retrieval Augmented Fine Tuning' or 'Retrieval Aware Fine-Tuning'?
Is it 'golden documents' or 'oracle documents'?
Is it 'sampled negative documents' or 'distractor documents'?

[3] Missing Implementation Details: The paper lacks detailed information on the choice of hyperparameters used in the main experiment (Table 2). This absence of details may hinder readers from reproducing the results presented in the paper.

---

> ### Author Rebuttal · Authors · 2024-05-31
>
> We would like to first thank the reviewer for their time and helpful feedback on our paper. We will address the reviewer’s comments in detail below.
>
> 1. Data Usage inconsistency: We want to clarify that in our experiment, we include the NQ [1] and TQA [2] datasets because they are relatively standard benchmarks for RAG (compared to PubMed and Huggingface). To study the effect of p% (percentage of the golden documents in the training dataset), we use the standard retriever trained in the Replug [3] paper and report the performance of RAFT on it. We will clarify this point in our next revision.
>
> 2. Term inconsistency: We apologize for using different terms. The terms should be “Retriever Augmented Fine-Tuning”, “Golden Documents”, and “Sampled negative documents”. We’ll update the paper with consistent naming.
>
> 3. Implementation details: We thank the reviewer for highlighting the missing details. We use a standard supervised fine-tuning method from the FastChat repository [4], with a learning rate of 1e-5, a warmup ratio of 0.03%, a batch size of 64, a total epoch of 2, and a percentage of golden documents of 80%. These details will be updated in the next revision and the code will be open-sourced.
>
> [1] Shi, Weijia, et al. "Replug: Retrieval-augmented black-box language models." arXiv preprint arXiv:2301.12652 (2023).
> [2] Kwiatkowski, Tom, et al. "Natural questions: a benchmark for question answering research." Transactions of the Association for Computational Linguistics 7 (2019): 453-466
> [3] Joshi, Mandar, et al. "Triviaqa: A large scale distantly supervised challenge dataset for reading comprehension." arXiv preprint arXiv:1705.03551 (2017)
> [4] Zheng, Lianmin, et al. "Judging llm-as-a-judge with mt-bench and chatbot arena." Advances in Neural Information Processing Systems 36 (2024)

---

### Decision · Program_Chairs · 2024-07-10

**Decision:**

Accept

**Comment:**

The paper studies domain-specific RAG fine-tuning for both language and code. It shows convincing improvements from domain-specific fine-tuning with retrieved documents from the domain in the context of a language model, and also shows the effect of chain-of-thought style answers and the addition of irrelevant documents in the context. The components of the approach are ablated well.

Pros
* The paper is very clear and easy to understand
* The experiments are thorough and present convincing improvements, with additional experiments added during author response
* The evaluation spans different types of tasks, including language and code
* The results will be of interest to researchers/practitioners

Cons
* The method is not very novel, with very similar methods previously applied to more general domain RAG fine-tuning
* It would be nice to evaluate the reasoning chains of the model – e.g. how often are references to retrieved paragraphs in the answer actually correct